# Effects of Self RehAbilitation Video Exercises (SAVE) on Functional Restorations in Patients with Subacute Stroke

**DOI:** 10.3390/healthcare9050565

**Published:** 2021-05-11

**Authors:** Seung-Hwan Jung, Eunhee Park, Ju-Hyun Kim, Bi-Ang Park, Ja-Won Yu, Ae-Ryoung Kim, Tae-Du Jung

**Affiliations:** 1Department of Rehabilitation Medicine, Kyungpook National University Hospital, Daegu 41944, Korea; pyromyth@naver.com (S.-H.J.); kjoohyun88@gmail.com (J.-H.K.); ryoung20@hanmail.net (A.-R.K.); 2Department of Rehabilitation Medicine, School of Medicine, Kyungpook National University, Daegu 41944, Korea; ehmdpark@knu.ac.kr; 3Department of Rehabilitation Medicine, Kyungpook National University Chilgok Hospital, Daegu 41404, Korea; 4Unit of Rehabilitation Therapy, Kyungpook National University Chilgok Hospital, Daegu 41404, Korea; biang8@naver.com (B.-A.P.); jawon0809@naver.com (J.-W.Y.)

**Keywords:** stroke recovery, rehabilitation, physiotherapy, exercise delivery

## Abstract

*Background*: Additional exercise therapy has been shown to positively affect acute stroke rehabilitation, which requires an effective method to deliver increased exercise. In this study, we designed a 4-week caregiver-supervised self-exercise program with videos, named “Self rehAbilitation Video Exercises (SAVE)”, to improve the functional outcomes and facilitate early recovery by increasing the continuity of rehabilitation therapy after acute stroke. *Methods*: This study is a non-randomized trial. Eighty-eight patients were included in an intervention group (SAVE group), who received conventional rehabilitation therapies and an additional self-rehabilitation session by watching bedside exercise videos and continued their own exercises in their rooms for 60 min every day for 4 weeks. Ninety-six patients were included in a control group, who received only conventional rehabilitation therapies. After 4 weeks of hospitalization, both groups assessed several outcome measurements, including the Berg Balance Scale (BBS), Modified Barthel Index (MBI), physical component summary (PCS) and the mental component summary of the Short-Form Survey 36 (SF-36), Mini-Mental State Examination, and Beck Depression Inventory. *Results*: Differences in BBS, MBI, and PCS components in SF-36 were more statistically significant in the SAVE group than that in the control group (*p* < 0.05). Patients in the SAVE group showed more significant improvement in BBS, MBI, and PCS components in SF-36 as compared to that in the control group. *Conclusions*: This evidence-based SAVE intervention can optimize patient recovery after a subacute stroke while keeping the available resources in mind.

## 1. Introduction

Stroke survivors are widely known to be easily deconditioned and hardly recover their abilities to perform activities of daily living (ADLs) in a short period. Stroke remains a leading cause of disability worldwide [1], with persistent neurological impairments which lead to limited activities and serious deconditioning [2,3]. At post-stroke, the most common and widely recognized impairment is motor weakness, and thereby, stroke rehabilitation should focus on the recovery of movement impairments and related functions. Moreover, patients who sustain severe strokes such as intracranial hemorrhage or ischemic brain injury are often extremely physically deconditioned, primarily due to prolonged physical inactivity. Therefore, setting goals and physical or occupational therapies are considered as an essential point in the stroke rehabilitation field even though better therapeutic options or appropriate duration remains debatable.

It is important for stroke survivors to start rehabilitation treatment as early as possible [4,5]. Rønning reported that hospital-based rehabilitation in subacute phase stroke survivors revealed significantly improved outcome measures rather than in the chronic phase [6]. Since adherence to physical activities is critical to improve post-stroke rehabilitation, patients should perform frequent and continuous exercises in their daily life [7]. In addition, a dose-dependent effect of rehabilitation was observed on the functional improvement of patients with stroke for the first 6 months post-stroke [8,9]. To increase patient adherence and self-efficacy and to facilitate telerehabilitation and remote monitoring by the rehabilitation team [10], exercise with e-health methods and combined with telerehabilitation services supported patient- and caregiver-mediated exercises was developed [11,12,13]. There have been several reported pieces of research regarding additional exercise [8,9,14], caregiver-mediated exercise delivery [12,15], exercise with e-health application [16], and systematic reviews or meta-analysis [17,18,19]. Galvin [17] reported that additional exercise therapy on the lower extremity showed significant improvement in walking speed. Huang [17] revealed a dose-dependent effect of additional rehabilitation exercises on functional improvement in subacute stroke survivors. Those researches have revealed that increased intensity and adherence of rehabilitation exercises leads to better physical and functional recovery. The advantages of these systems are their cost-effectiveness and easy communication with patients to allow continuous rehabilitation, whereas their disadvantages included the need for optimized rehabilitation programs according to the type of disease, which is sometimes impossible due to high costs [11,13]. Furthermore, considering that most patients with stroke and caregivers are aged people, they may find it difficult to use specific electronic devices to perform sufficient exercises. Rehabilitation treatment for subacute stroke patients is related to cost concerns. Alternative methods are needed to increase the amount and continuity of exercise therapy without increasing healthcare costs with easier accessibility [20,21]. Self exercise, i.e., patients perform the exercise themselves, would be a promising choice.

To date, researchers have dealt with the novelty of increasing the duration and amount of exercise therapy. We developed this Self Rehabilitation Video Exercises (SAVE) program as a novel strategy to deliver exercise efficiently in subacute stroke survivors. This program consists of 28 videos of the therapist demonstrating self-exercises. Using those patient-tailored videos, therapists educated patient-caregiver couples to perform exercises with their full responsibility. Instead of teaching one of the family members [12] or using telerehabilitation services [16], we encouraged both patient-caregiver couples to participate in this program. Patients received rehabilitation treatments and continued their exercises at the bedside, while their caregivers were educated on how to supervise and support them. We also provided exercise videos providing proper visual guidance, which enable patients to review their therapy sessions and continue unceasing rehabilitation treatment. The intervention group performed a patient- and caregiver-mediated exercise program in addition to conventional rehabilitation therapy compared with the control group who received conventional rehabilitation therapy alone. We evaluated the effectiveness of the additional SAVE program to improve functioning, disability, and health framework according to WHO’s international classification after stroke compared with the conventional rehabilitation in subacute stroke patients.

## 2. Materials and Methods

### 2.1. Participants

This study is a non-randomized trial. The inclusion criteria were as follows: (1) patients were diagnosed with the first-ever stroke with brain computed tomography or magnetic resonance imaging; (2) patients were transferred to the Department of Rehabilitation Medicine after 4–6 weeks after stroke; (3) patients were aged ≥ 18 years old; (4) patients were vitally stable and kept in good general medical conditions to participate inpatient rehabilitation; (5) patients should communicate with our rehabilitation team or physicians and obey verbal commands; (6) a caregiver who is willing to participate in the program were also eligible. The exclusion criteria were as follows: (1) patients previously suffered from head trauma, space-occupying lesions, cerebral metastatic disease, or central nervous system infections; (2) patients had impaired cognition or vision loss or aphasia; (3) patients were non-ambulatory before the stroke; (4) patients were diagnosed with osteoporosis or compression fracture; (5) patients had severe musculoskeletal problems such as adhesive capsulitis, knee osteoarthritis.

In the intervention group, a total of 227 patients met the inclusion criteria who were admitted to Kyungpook National University Chilgok Hospital between October 2016 and December 2019. One hundred thirty-nine patients gave up intervention due to their developing secondary comorbidities (ex. pneumonia, urinary tract infection) or poor compliance. Finally, we included 88 patients as the intervention group (SAVE group). On the other hand, the control group was a historical cohort and included 96 patients who received the conventional rehabilitation therapy during inpatient in the Department of Rehabilitation Medicine from January 2015 to August 2016 (Figure 1).

All patients generally received conventional rehabilitation therapy consisting of physical therapy (PT) and occupational therapy (OT). PT included lower extremities strengthening and range-of-motion exercises, sitting and standing balance training, stance and weight transfer training, and gait training. It also included visual feedback or eye and body coordination when the patient complained of sensory function impairment. OT included upper extremities strengthening and range-of-motion exercises, hand fine motor training, in-hand manipulation training, and activity training for daily living. It also included cognitive function improvement training and swallowing exercises. PT and OT sessions proceeded two times a day, with each therapy taking 30 min per session for 4 weeks. Interventions were largely derived from the Guide of Physical Therapist Practice [22]; therapeutic components included both function-focus activities such as ambulatory training or ADL training, and impairment-focus activities such as facilitation technique or perceptual-motor training. Every patient was assigned to physical and occupational therapists to perform one-to-one therapy during whole sessions in the rehabilitation unit. This study was approved by the Institutional Review Board of Kyungpook National University Chilgok Hospital (No. 2018-08-011-002).

### 2.2. Self Rehabilitation Video Exercises Program

The content of the SAVE program consisted of 4-weeks of exercise therapy, executed with a caregiver, in addition to the conventional rehabilitation therapy. Harris et al. reported that only a 4-week additional self-administered exercise program resulted in greater improvement in upper limb function in subacute stroke patients [23]. Galvin et al. designed an 8-week additional exercise program and reported significant improvement in functional recovery [12]. Instead of designing an 8-week program, we designed a 4-week additional intervention program with nearly doubled exercise time, 35 min [12] to 60 min. The SAVE program was facilitated by a trained physical therapist during weekly sessions. The therapist could choose four exercise program categories, consisting of 28 videos aimed at improving strength, balance, and mobility (Figure 2). Each video-recorded therapist demonstrated proper body position and exercise in detail. We inserted Korean subtitles in every video to deliver content accurately (Figure 3).

For each patient, exercises were blended into a patient-tailored, customized training regimen associated with patient goals. Patient–caregiver couples were inspired to contact their therapist after daily exercise sessions. After the first therapy sessions, physicians and therapists held team conferences and selected a patient-tailored set of exercises for each patient. Patients and their caregivers watched the selected exercise videos and were trained to perform the selected exercises at least five times a week for 60 min when they returned to their ward. We sent those selected videos to patients’ or caregivers’ smartphones, and they downloaded and watched those videos and exercised as instructed. Before starting every therapeutic session, the therapist checked and provided feedback on their exercises. Every week, or when the patient achieved their goals, the therapist assigned next-level exercises depending on the patient’s progress. This meant that patients performed 20 h of caregiver-mediated exercises in addition to conventional rehabilitation therapies during the 4-week intervention period. All physical and occupational therapists were thoroughly trained following a training course.

Participants in the control group received conventional rehabilitation therapies. Since the therapeutic sessions are designed according to patient goals, no restrictions were observed with respect to the content, time, or duration of the rehabilitation therapy. Task and context specificity are important aspects of PT after stroke. With this, current guidelines indicated that exercises to improve functional outcomes such as physical condition, standing or gait balance, and walking competence are recommended.

### 2.3. Outcome Measures

The primary outcome measurements used were Berg Balance Scale (BBS), Mini-Mental State Examination (MMSE), Modified Barthel Index (MBI), physical component summary (PCS), and the mental component summary (MCS) of the Short-Form Survey 36 (SF-36), and Beck Depression Inventory (BDI) according to the WHO’s International Classification of Functioning, Disability, and Health (ICF).

First, the functioning domain was assessed using BBS [24] and MMSE [25]. The BBS consists of a five-point ordinal scale ranging from 0 to 4 in 14 static and dynamic balance items [24]. In addition, the MMSE is a 30-point questionnaire that is widely used in clinical settings to measure cognitive impairment. We used the Korean version of the Mini-Mental State Examination, K-MMSE [25].

Second, the disability domain was assessed using MBI [26]. The MBI is an ordinal measure of performance in activities of daily living (ADL). We used the Korean version of the MBI, whose reliability and validity has been proven [27].

Third, the health domain was assessed using the PCS and MCS of the SF-36, and the Beck Depression Inventory (BDI). The SF-36 is a widely used and patient-reported measure of health status [28]. PCS comprises four physical domain subscales: physical functioning, role functioning-physical, bodily pain, and general health. In addition, MCS has four mental domain subscales: vitality, social functioning, role functioning-emotional, and mental health. We used the Korean version of the SF-36 for its reliability and validity, which has been proven. Moreover, BDI is a 21-questionnaire that is widely used to measure the severity of depression. We used the Korean version of the BDI [29].

These outcomes were administered at baseline (T_0_) and post-intervention (T_1_) at 4-week follow-up. All the measurements were being assessed by expert physiotherapists who were specialized in assessments and physical evaluation. Every patient underwent those assessments every 4 weeks by the experts, and those experts were not informed whether the patient is under certain research groups or not. All assessments were completed using a standardized assessment kit. We compared T_0_ and T_1_ in each group using the paired *t*-test. Changed values, defined by the change from baseline to post-intervention, were compared between the two groups using the Student’s *t*-test. We used SPSS version 15.0 (SPSS Inc., Chicago, IL, USA).

## 3. Results

Demographic characteristics showed no significant differences between the two groups at baseline (T_0_) (Table 1). For within-group comparisons, both groups showed significant improvement in BBS, MBI, and PCS of the SF-36 component and MMSE scores after the 4-week rehabilitation therapy (Table 2). Comparing the initial and post-intervention outcome measurements, the changed values were defined by the change from pre- to post-intervention from each group, and the SAVE group showed significant improvement in BBS, MBI, and PCS of the SF-36 scores compared to the control group after the 4-week interventional program (Table 3). Primary outcome measures about physical ability, BBS, and PCS of SF-36 component showed a significant difference as compared to the control group. Moreover, MBI, which is about the functional ability of a patient, also showed a significant difference due to improved physical abilities.

## 4. Discussion

In this study, both groups showed significant improvement in BBS, MBI scores after the 4-week rehabilitation therapy. Moreover, in the SAVE group of individuals with acute stroke, additional video-based self-mediated therapy significantly improved the physical ability, balance, and functional ADLs as compared to the conventional rehabilitation therapy alone. The SAVE intervention resulted in about 12-point improvement in BBS score. Lower limb strengthening exercise [24] and core strengthening exercise [25,26] were responsible for significant BBS improvement in subacute stroke survivors. A change of BBS points more than six is considered to be a significant improvement in acute stroke rehabilitation [27,28]. The significant difference in exercise time by oneself or with a caregiver might explain these significant treatment effects. Reviewing every session in person and practicing together with a caregiver led to a positive effect on the physical functioning of patients. These differences in efficacy were most evident in post-intervention and highly improved functional states under the same hospitalization period that may facilitate the transition from the inpatient rehabilitation center to the home environment, which will reduce further expenses. Although Vloothuis et al. reported no differential effects with respect to functional outcome measures [13], our results are compatible with that of previous studies in which a significant improvement was shown in participants after an additional focused exercise therapeutic program [29,30].

However, we found no differential effect with regard to outcome measures of psychosocial components such as MMSE, BDI, and MCS of SF-36. According to previous studies, the incidence of anxiety in patients with stroke is significantly higher than that in healthy age-matched controls [31], and caregiver-mediated exercises seem to have a positive effect on the quality of life of patients [13]. Depression, psychosocial stress, and anxiety are major symptoms in stroke survivors [32] and may persist for years [33,34,35]. The psychosocial components significantly deteriorate quality of life for a prolonged period [34,36]. This might explain the lack of effects found on psychosocial stress and anxiety measures in this study [37,38].

In addition, post-stroke fatigue can also explain the lack of effects on psychosocial outcome measures. Fatigue is a common and major problem after stroke, with previously reported 38% to 77% ranging frequencies [39], considered as one of the worst sequelae of stroke [40] and has an enervating influence on ADL [41,42]. Fatigue is an independent factor concerned with health-related quality of life in patients with stroke [42], and reported levels of post-stroke fatigue are high and remain fairly stable over time. According to previous studies, a significant relationship was found between depression and post-stroke fatigue both in cross-sectional [40,43,44] and longitudinal analyses [45,46,47,48]. Fatigue levels remaining stable over time in post-stroke patients [49] also explain the lack of effects on psychosocial outcome measures in this study.

This study has several limitations. First, we intended 1200 min of additional exercise time by the patient–caregiver couples; however, some patients in the control group also reported additional exercise time by themselves or performed exercises with a caregiver. Therefore, there might have been insufficient treatment in improving other functional outcome measures. This type of contamination is often observed in stroke rehabilitation trials which require a long-term period of recruitment of several years to conclude [13,50,51]. In addition, we provided an exercise diary to participants and encouraged them to record their daily achievements. The therapist in charge checked the progress prior to the treatment session every day, but we did not investigate the achievement rate of participants separately.

Second, the incidence of depression in stroke patients’ caregivers in previous studies [52] was significantly higher than that in healthy controls, and anxiety symptoms are predictors of long-term burden and emotional problems among caregivers [53]. Kalra et al. [54] suggested that the use of a well-structured program of activities under supervision by professional rehabilitation teams during the hospitalization period may serve to authorize consenting informal caregivers in their future role by instructing them in relevant skills. Mant et al. [55] also reported that additional family support after stroke significantly boosted social activities and also improved the quality of life of caregivers. In this study, we only focused on the rehabilitation of patients with stroke, and further studies should be conducted to follow-up patients with stroke and their caregivers.

Third, this study was a non-randomized trial that did not exclude the risk of selection bias and performance bias. Finally, since all the patients were able to follow our instructions, eligible participants with better compliance were included. This biased distribution might affect better outcome measurements. Future trials should take account of a randomized controlled design with a longer-term follow-up. Ultimately, inclusion and clinical assessments should include patients with acute stroke and also their caregivers.

## 5. Conclusions

Additionally, SAVE showed significant improvement in physical and functional abilities compared to only conventionally inpatient rehabilitation therapy in subacute stroke patients. This SAVE program which is designed to deliver patient-tailored exercise, can optimize patient recovery after subacute stroke while keeping the available resources in mind. In this study, the SAVE program facilitated unceasing rehabilitative exercises with higher patient adherence, which is important in post-stroke recovery. The most important thing is the incorporation of exercise into the patient’s daily life in order to maintain their therapy until the next session, thereby erasing boundaries between exercise and daily life.

## Figures and Tables

**Figure 1 healthcare-09-00565-f001:**
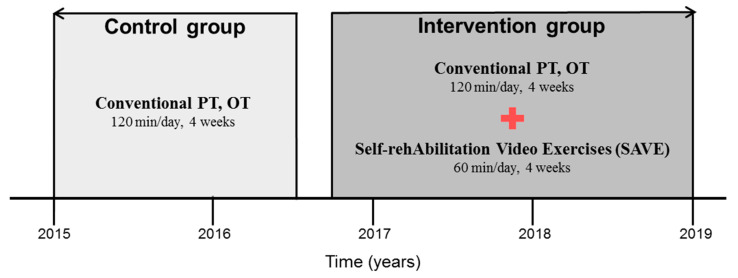
The timeline of enrolment of participants. OT: occupational therapy; PT: physical therapy.

**Figure 2 healthcare-09-00565-f002:**
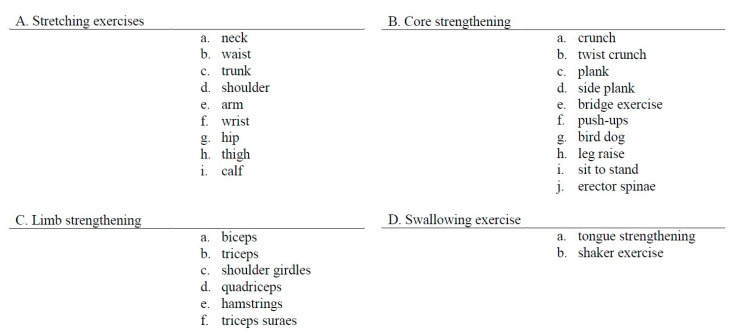
Self Rehabilitation Video Exercises video programs.

**Figure 3 healthcare-09-00565-f003:**
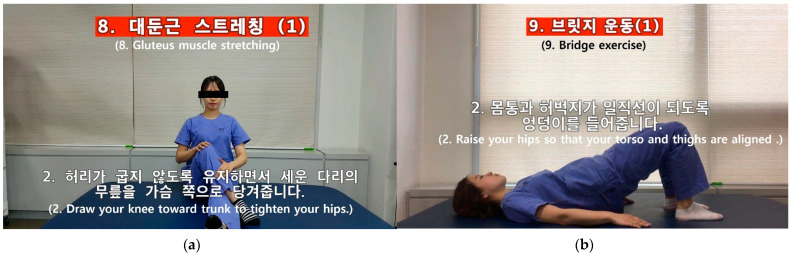
Self Rehabilitation Video Exercises video program. (**a**) Hip muscle stretching exercise; (**b**) bridge exercise.

**Table 1 healthcare-09-00565-t001:** Clinical characteristics in control and SAVE groups at baseline (T_0_).

Baseline Variables	Control (*N* = 96)	SAVE (*N* = 88)	*p*-Value
Sex			0.949
Male, N (%)	53 (55%)	49 (56%)	
Female, N (%)	43 (45%)	39 (44%)	
Age	61.9 ± 17.3	60.8 ± 14.8	0.644
Duration from onset (Days)	44.95 ± 10.84	46.40 ± 11.10	0.540
BBS	26.64 ± 20.79	28.05 ± 18.31	0.498
MBI	49.87 ± 29.43	50.46 ± 24.15	0.835
BDI	14.45 ± 11.60	12.08 ± 10.30	0.082
MMSE	22.88 ± 7.84	23.86 ± 6.15	0.193
SF-36			
MCS	29.83 ± 10.41	30.02 ± 11.06	0.376
PCS	35.56 ± 11.84	34.28 ± 9.48	0.890

N, the number of patients; mean ± SD, mean ± standard deviation. BBS, Berg Balance Scale; BDI, Beck Depression Inventory; MBI, Modified Barthel Index; MMSE, Mini-mental State Examination; MCS, Mental Component Summary of the Short Form (SF)-36 health survey; PCS, Physical Component Summary of SF-36 health survey.

**Table 2 healthcare-09-00565-t002:** Outcome measures in balance function, activities of daily life, the severity of depressive mood, and cognitive function at baseline (T_0_) and post-intervention at 4-week follow up (T_1_).

Outcome Measure	Control (*N* = 96)		*p*-Value (95% CI)	SAVE (*N* = 88)		*p*-Value (95% CI)
	T_0_	T_1_		T_0_	T_1_	
BBS	26.64 ± 20.79	33.56 ± 20.04	0.000 *(−11.92 to −7.36)	28.05 ± 18.31	39.88 ± 16.70	0.000 *(−14.48 to −9.73)
MBI	49.87 ± 29.43	59.24 ± 26.79	0.000 *(−14.00 to −9.08)	50.46 ± 24.15	66.45 ± 23.03	0.000 *(−22.66 to −15.41)
BDI	14.45 ± 11.60	14.73 ± 12.18	0.157 (−6.32 to 0.86)	12.08 ± 10.30	9.20 ± 9.18	0.087 (−2.21 to 4.31)
MMSE	22.88 ± 7.84	25.26 ± 6.48	0.000 *(−3.39 to −1.85)	23.86 ± 6.15	25.76 ± 5.12	0.000 *(−2.88 to −1.29)
PCS	29.83 ± 10.41	32.29 ± 10.93	0.026 *(−7.70 to −0.51)	30.02 ± 11.06	36.40 ± 10.13	0.000 *(−19.98 to −6.02)
MCS	35.56 ± 11.84	37.32 ± 11.37	0.803 (−22.94 to 19.32)	34.28 ± 9.48	38.53 ± 10.19	0.074 (−9.39 to 0.45)

Each cell represents the mean ± standard deviation. * represents a significant difference within-group comparison between T_0_ and T_1_ (*p* < 0.05). BBS, Berg Balance Scale; BDI, Beck Depression Inventory; MBI, Modified Barthel Index; MMSE, Mini-mental State Examination; MCS, Mental component summary of the Short-Form Survey 36 (SF-36); PCS, Physical component summary of SF-36.

**Table 3 healthcare-09-00565-t003:** The raw and change values in balance function, activities of daily life, the severity of depressive mood, cognitive function, and quality of life at pre-intervention (T_0_) and post-intervention at 4-week follow up (T_1_).

Outcome Measure	Control (*N* = 96)			SAVE (*N* = 88)			*p*-Value (95% CI)
	T_0_	T_1_	∆ T_1_-T_0_	T_0_	T_1_	∆ T_1_-T_0_	
BBS	26.64 ± 20.79	33.56 ± 20.04	7.98 ± 10.47	28.05 ± 18.31	39.88 ± 16.70	12.12 ± 12.45	0.002 *(−6.76 to −1.52)
MBI	49.87 ± 29.43	59.24 ± 26.79	11.97 ± 17.20	50.46 ± 24.15	66.45 ± 23.03	11.80 ± 15.27	0.001 *(−8.35 to −2.12)
BDI	14.45 ± 11.60	14.73 ± 12.18	0.70 ± 8.08	12.08 ± 10.30	9.20 ± 9.18	2.80 ± 9.00	0.108 (−0.46 to 4.66)
MMSE	22.88 ± 7.84	25.26 ± 6.48	2.03 ± 1.91	23.86 ± 6.15	25.76 ± 5.12	3.63 ± 3.21	0.760 (−0.66 to 0.91)
PCS	29.83 ± 10.41	32.29 ± 10.93	2.50 ± 8.81	30.02 ± 11.06	36.40 ± 10.13	7.16 ± 10.24	0.013 *(−8.33 to −0.99)
MCS	35.56 ± 11.84	37.32 ± 11.37	4.42 ± 11.18	34.28 ± 9.48	38.53 ± 10.19	3.07 ± 12.70	0.561 (−3.23 to 5.93)

Each cell represents the mean ± standard deviation. * represents a significant difference between intervention and control group in ∆ T_1_-T_0_ (*p* < 0.05). BBS, Berg Balance Scale; BDI, Beck Depression Inventory; MBI, Modified Barthel Index; MMSE. Mini-mental State Examination; MCS, Mental component summary of the Short-Form Survey 36 (SF-36); PCS, Physical component summary of SF-36.

## Data Availability

Data sharing is not applicable to this article.

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
