# Peer review of "Effects of Self RehAbilitation Video Exercises (SAVE) on Functional Restorations in Patients with Subacute Stroke"

_healthcare, 2021, doi:10.3390/healthcare9050565_

Round 1

Reviewer 1 Report

I have attached a file, please check it out.

Reviewer 2 Report

Thank you for giving me the opportunity to review the manuscript. Please see my comments below:

Introduction

  1. Describe why rehabilitation in the subacute phase after stroke is so important. Describe the benefits you can expect from starting with the subacute phase rather than the chronic phase.
  2. Describe what is the difference between the previous studies' findings and the interventions conducted in the current study, what has been improved, and what can be expected. 
  3. (Page 2) What is the patient-mediated exercise? It does not make sense. Family- or caregiver-mediated exercise make sense. According to the reference you cited, those did not mean patient-mediated exercise. 
  4. Describe the research purpose clearly.

Materials and methods

  1. You used the term 'historical control,' which I guess it meant traditional control group. The traditional control group usually did not receive any intervention. 
  2. In your study, what is the specific conventional rehabilitation therapy? In page 4, you describe some but it was not enough. Which type of exercises, how often, how long a session, by how? Both groups received the conventional rehabilitation therapy but the therapy was not explained clearly.
  3. Page 2, Participants) You described you conducted a prospective non-randomized trials with historical controls. However, in page 3, you described that a historical group retrospectively included 88 patients from January 2015 to August 2016. Importantly, your study was approved in 2018. Can you make it clear?
  4. Intervention - why did you plan to provide 4 week intervention program? any evidence? which efforts did you make to retain your samples? Add the strategies in the intervention section.
  5. Outcome measures - Describe each measure clearly including developer, purpose, coding, interpretation, reliability/validity, etc.
  6. Add data analysis method. You only mentioned changed values. And technically, it was better to show to significant changes by paired t-tests between baseline and control in both groups and significant differences between the two post-groups. Also, Describe data analysis method you used to describe your purpose in your results.

Results

  1. Demographic characteristics: did you measure sex, disease duration (weeks), any underlying diseases? If you did so, add those things.
  2. In Table 1, MMSE scores of control group and intervention group are 22.88+-7.84, 23.86+-6.15, respectively. As you know, 20 to 24 suggests mild dementia and 13 to 20 moderate dementia. It seems like you included patients with moderate cognitive deficits. It is very important about human subject issue. You needed to receive informed consent from patients and caregivers...
  3. In table 2, add each sample size in control and intervention groups. 

Discussion

  1. Page 6, fist line. Your study did not make significant improvement in MMSE. And add MMSE in the first line of second paragraph in the discussion section. And add to describe why PCS was improved but MMSE was not improved, using the evidences.
  2. You did not ask or measure exercise time. How did you know?
  3. What is the hospital setting? Few acute tertiary hospitals are hospitalized in ward for 4 weeks for subacute stroke. Maybe it is only for your hospital....
  4. Your intervention effect could be overestimated if you excluded patients with poor compliance. Patients with good compliance would be more likely to show more significant improvement. How did you decide poor compliance? You may want to add it in the limitation.

Round 2

Reviewer 1 Report

All the points I made have been revised, so I have no comments to author.

Author Response

Reviewer 1 remarks "All the points have been revised" and has no comments to author.

We have changed minor typographical errors:

(Line 181) Figure 1 → Figure 2

(Line 183) Figure 2 → Figure 3

Thank you again for your detailed opinions about the manuscript. With your sincere advice, this study conveys the contents more clearly. 

Reviewer 2 Report

Overall my comments were addressed in the revised manuscript. Just consider the minor two comments below:

  1. Provide n(%) for sex in Table 1.
  2. Replace SAVE in Table 2 with intervention in order to be consistent with the intervention in Table 1.

Author Response

Also, we have changed minor typographical errors:

(Line 181) Figure 1 → Figure 2

(Line 183) Figure 2 → Figure 3
